# Modopathies Caused by Mutations in Genes Encoding for Mitochondrial RNA Modifying Enzymes: Molecular Mechanisms and Yeast Disease Models

**DOI:** 10.3390/ijms24032178

**Published:** 2023-01-22

**Authors:** Martina Magistrati, Alexandru Ionut Gilea, Camilla Ceccatelli Berti, Enrico Baruffini, Cristina Dallabona

**Affiliations:** 1Department of Chemistry, Life Sciences and Environmental Sustainability, University of Parma, Parco Area delle Scienze 11/A, 43124 Parma, Italy; 2Department of Medicine and Surgery, University of Parma, Via Gramsci 14, 43126 Parma, Italy

**Keywords:** mt-RNA modifying enzymes, mitochondrial modopathies, novel mutations, yeast model

## Abstract

In eukaryotes, mitochondrial RNAs (mt-tRNAs and mt-rRNAs) are subject to specific nucleotide modifications, which are critical for distinct functions linked to the synthesis of mitochondrial proteins encoded by mitochondrial genes, and thus for oxidative phosphorylation. In recent years, mutations in genes encoding for mt-RNAs modifying enzymes have been identified as being causative of primary mitochondrial diseases, which have been called modopathies. These latter pathologies can be caused by mutations in genes involved in the modification either of tRNAs or of rRNAs, resulting in the absence of/decrease in a specific nucleotide modification and thus on the impairment of the efficiency or the accuracy of the mitochondrial protein synthesis. Most of these mutations are sporadic or private, thus it is fundamental that their pathogenicity is confirmed through the use of a model system. This review will focus on the activity of genes that, when mutated, are associated with modopathies, on the molecular mechanisms through which the enzymes introduce the nucleotide modifications, on the pathological phenotypes associated with mutations in these genes and on the contribution of the yeast *Saccharomyces cerevisiae* to confirming the pathogenicity of novel mutations and, in some cases, for defining the molecular defects.

## 1. Introduction

In most eukaryotes, mitochondria are the organelles in which most of the ATP is synthesized. Mitochondria contain their own genome in multiple copies, which is called mitochondrial DNA, or mtDNA. Although in different organisms the length of the mtDNA and the number of mitochondrial genes is different, in most cases mtDNA encodes for all the mitochondrial tRNAs (mt-tRNAs), the mitochondrial rRNAs (mt-rRNAs) and for a few proteins which make part of the respiratory complexes. Most of the mitochondrial proteins, including those involved in oxidative phosphorylation (OXPHOS), are encoded by nuclear genes and transported into mitochondria. However, the few protein-encoding genes present in mtDNA are transcribed and translated inside the organelle thanks to its transcription and translation machinery.

In mammals, mtDNA contains 22 genes for mt-tRNA, two genes for mt-rRNA and 13 genes for subunits of the respiratory complexes. Since all these subunits are crucial for the OXPHOS activity, mutations in mtDNA which affect mitochondrial translation are associated with mitochondrial diseases (MDs), which have a maternal inheritance, since the mtDNA is inherited from the mother. To date, more than 300 pathological mutations have been identified in genes encoding for mt-tRNAs and mt-rRNAs, which account for approximately 50% of the mitochondrial pathological mutations [1,2]. These mutations are associated with a plethora of different mitochondrial diseases [3], which occur when the number of mtDNA mutant copies exceeds the threshold in an affected tissue [4].

MDs are caused also by mutations in nuclear genes encoding, in most cases, mitochondrial proteins. Approximately 300 genes have been associated with several MDs, which are inherited according to Mendelian laws, as autosomal recessive (AR), autosomal dominant (AD), X-linked recessive (XR), X-linked dominant (XD) or are de novo mutations [5]. In the last decade, a novel class of MDs has been defined as mitochondrial modopathies [6], which are diseases caused by the lack of or the decrease in specific nucleotide modifications in mt-tRNA or mt-rRNA molecules which affect mitochondrial translation. In general, modopathies can be caused by altered or a lack of modifications in cytoplasmic RNA, mitochondrial RNA, or both [7].

Most modopathies are caused by a lack of modification of one or more tRNAs. All tRNA molecules have a highly conserved secondary structure similar to a cloverleaf, which consists of stems and loops: acceptor stem, dihydrouridine (D) loop, anticodon loop, variable loop and thymidine-pseudouridine-cytidine, or TΨC (T), loop. Thanks to the base pairing in loops, tRNA form an L-shaped tertiary structure via multiple hydrogen bonds between loops and helices, characterized by an anticodon arm/D arm and a T arm/acceptor stem. All tRNAs possess several and specific modifications, which are post-transcriptionally added to tRNA by specific modifying enzymes. To date, 43 types of tRNA modifications have been identified in humans, which are introduced by 73 enzymes and accessory proteins [7]. At least 50 genes encoding for such proteins are associated with diseases when mutated. Regarding mt-tRNAs, at least 18 types of nucleotide modifications have been identified in humans, as reported in Table 1.

Overall, tRNA modifications have several functions. Most of the tRNA modifications regulate the local structure and stabilize the global structure of each tRNA, allowing the recognition of their interactors, in particular the specific aminoacyl-tRNA synthetase and the mRNA-ribosome complex. Some modifications avoid tRNAs being degraded by specific endonucleases, increasing their stability. Modifications in nucleotides of the anticodon are critical in order to recognize multiple synonymous codons for the specific amino acid and to avoid mismatch pairing with wrong codons. Most of the single human mt-tRNAs must recognize all the synonymous codons, up to four for some mt-tRNAs. Most synonymous codons diverge for the third base, which is called the wobble position, which is recognized by the base at position 34, the first base of the anticodon, forming the so-called wobble pair. Modifications at position 34 have the function to relax the recognition of the wobble position, with a limit. If an amino acid is encoded by four synonymous codons in which the wobble position is occupied by each of the four bases (for example, GGN for glycine), the anticodon must pair with each of the codons, that is the first base of the anticodon should pair with each of the four bases. On the contrary, if an amino acid is encoded by two synonymous codons in which the wobble position is occupied by just a pyrimidine (for example, GAY for aspartate), the first base of the anticodon must pair with U and C, but not with purines, and vice-versa. For such reasons, position 34 in the tRNAs is the most modified base considering the whole group of tRNAs. Furthermore, position 37, adjacent to the third base of the anticodon, is strongly modified in most tRNA, especially with bulky groups. Modifications in such a position can have several functions: stabilization of codon–anticodon pairing, maintenance of the reading frame in order to avoid frameshift and/or preventing unwanted base pairing within the anticodon loop. Other modifications have specific roles: for example, some modifications serve as recognition for specific modifying enzymes which introduce the second modification in adjacent bases (reviewed in [7,56,57,58,59,60]).

In humans, at least 21 genes encode for mt-tRNA modifying enzymes, whereas, for some other genes, their involvement is just putative. As reported in Table 2, mutations in 10 of these genes are associated with modopathies, with different characteristics concerning the age of onset, the tissues involved, the kind of inheritance, the symptomatology and the prognosis (reviewed in [2,7,13,59,60,61,62]).

A second, minor group of modopathies are caused by mutations in genes encoding for enzymes involved in the modification of mitochondrial rRNAs (mt-rRNAs) [13,59]. Human mtDNA encodes for two rRNA: the 12S and the 16S mt-rRNA. The former is a component of the 28S small subunit (mtSSU), the latter of the 39S large subunit. The two subunits are the components of the 55S mitochondrial ribosome, which is composed of 82 nuclear-encoded proteins, two mitochondrial-encoded rRNA and one mitochondrial-encoded tRNA (mt-tRNA^Val^ or mt-tRNA^Phe^), which has the role of the 5S rRNA in cytoplasmic ribosomes [94]. However, it has been demonstrated that in humans the 5S RNA subunit of the cytoplasmic ribosomes is imported, through a complex mechanism that involves several proteins, into the mitochondria, where it makes part of the mitoribosomes, increasing the translation efficiency [95,96,97,98,99]. Both 12S and 16S mt-rRNA molecules, but not the imported 5S rRNA, undergo nucleotide modifications, although to a lesser extent than their cytoplasmic counterparts [100,101,102]. Modifications in 10 positions are known, as reported in Table 3.

Modifications of mt-rRNAs have several functions, including rRNA maturation and stability, the assembly of ribosomal subunits, the optimization of translation and, concerning the modifications of ribose, the stabilization of the structure of the ribosomal RNA-loop as well as the optimization of the peptidyl transferase reaction [13,59,105,111,112,113].

Whereas a polymorphism in *METTL15* is associated with obesity [114] and a polymorphism in *TFB1M* is associated with type 2 diabetes [115], only a single primary MD is associated with mutations in the *MRM2* gene involved in mt-rRNA modification: mitochondrial DNA depletion syndrome 17 (OMIM #618567) with infancy onset [116].

It must be underlined that, besides covalent modifications of mtRNAs catalyzed by modifying enzymes, mtRNAs undergo non-covalent interactions with other proteins or molecules such as iron-containing groups and polyamines which are critical for their function or the function of their interactors, such as heme bioavailability [117,118,119]. However, in this review, only modopathies caused by mutations in mt-tRNA and mt-rRNA modifying enzymes will be discussed.

The yeast *Saccharomyces cerevisiae* has been intensively used as a model system to confirm the pathogenicity and the molecular and phenotypic consequences of mutations in genes associated with MDs, either mitochondrial or nuclear (reviewed in [120,121,122,123,124]. The most important characteristic for such studies is that *S. cerevisiae* is able to grow either on fermentable or non-fermentable (or oxidative) carbon sources. If mitochondrial oxidative phosphorylation (OXPHOS) is impaired, yeast has a limited capacity or is unable to perform respiration and to grow on non-fermentable carbon sources. Since most MDs result in an OXPHOS deficiency, the pathogenicity of novel putative mutations, in genes already known to be associated with an MD or in novel genes, can be assessed primarily by measuring the growth on non-fermentable carbon sources and the oxygen consumption rate. Several yeast models have been constructed to validate such mutations. These models are mainly based on the gene disruption of the yeast ortholog of the human gene and on complementation studies with wild type and mutant alleles. Depending on specific genes, these assays are performed through homologous complementation with the yeast gene carrying a mutation in the equivalent position of the human ortholog, heterologous complementation with the human gene carrying the mutation or complementation with a mutant chimeric construct harboring a fragment of the yeast gene and a fragment of the human gene [124].

Yeast mtDNA, which is longer compared with its mammalian counterpart and can be 68 Kbp (in short strains) to 86 Kbp (in long strains), contains seven genes encoding for subunits of the respiratory complexes (*COB* for complex III, *COX1*, *COX2*, *COX3* for complex IV, *ATP6*, *ATP8* and *ATP9* for complex V), one gene, *VAR1*, encoding for a subunit of the mitochondrial ribosome; one gene, *RPM1*, encoding an RNA subunit of the RNase P, which is involved in the processing of the mitochondrial pre-tRNAs; 24 genes for tRNAs; and two genes for the 15S rRNA and for the 21S rRNA. In addition, some genes encoding for maturases, endonucleases and a reverse transcriptase are present inside the introns of the *COX1*, *COB* and *21S rRNA* genes.

Yeast possesses two more mt-tRNA genes compared with mammals for two main reasons. First, yeast and mammals have two different mitochondrial genetic codes. In humans, the codons AGR are stop codons, whereas in yeast they encode for arginine; in humans, the codons CUN encode for leucine, in yeast they encode for threonine. In addition, yeast possesses two mt-tRNA^Met^, one for initiation and one for the elongation. Human and yeast mt-tRNAs are reported in Table 4.

In yeast, 15S rRNA, orthologous to the human 12S rRNA, makes part of the small ribosomal subunit, whereas 21S rRNA, orthologous to the 16S rRNA, makes part of the large ribosomal subunit.

Thanks to the similarity between yeast and humans regarding mtDNA replication and mtDNA stability, transmission, transcription and translation, *S. cerevisiae* has been used to validate mutations in genes encoding for enzymes or other proteins involved directly or indirectly in such pathways, as reviewed in [124,125,126,127,128]. Mutations in mitochondrial and nuclear genes involved in mitochondrial translation can result in an impairment of mitochondrial protein synthesis, which can be evaluated by in vivo or in organello radioactive protein labeling assay followed by SDS-PAGE and a measurement of the radioactive signal of each of the eight proteins encoded by mtDNA [129].

Several of the mt-RNA modifying enzymes found in humans are present and conserved in yeast too, especially the enzymes which modify the tRNAs inside or near the anticodon. A limited number of mt-rRNA modifying enzymes are also present. In this review, we will report the characteristics of such enzymes, the phenotypes caused by mutations associated with genes encoding for these enzymes and the use of *S. cerevisiae* as a model for the evaluation of the pathogenicity of human mutations. The advantages and limits of this model will be also discussed.

## 2. Characteristics of tRNAs Modifying Enzymes Associated with Modopathies and Characterization of the Mutant Variants in Yeast

### 2.1. Mutations in the Yeast 15S mt-rRNA Which Abolish a Base Pair Affect the Phenotype Associated with Mutations in Specific Genes Encoding for mt-tRNA Modifying Enzymes

In *S. cerevisiae*, the 15S mt-rRNA contains a cytosine at position 1477 and a guanosine at position 1583 that interact forming a base pair. This base-pairing, together with the surrounding nucleotides, forms a subsite inside the A-site which can be bound by aminoglycosides such as paromomycin, an antibiotic that inhibits at high concentrations the respiratory growth of *S. cerevisiae*. Indeed, aminoglycosides can bind to the mt-rRNA A-site, thus inhibiting the elongation by misreading or premature termination and preventing mitochondrial protein synthesis [130]. If a C to G transversion occurs at position 1477, the base pair with G1583 is prevented, leading to the inability of paromomycin to bind the A-site and thus to paromomycin resistance (P^R^) [24,27,131,132]. On the contrary, in humans, the mitochondrial 12S mt-rRNA contains, at the equivalent positions of the yeast 15S mt-rRNA, C1494 and A1555, thus preventing the formation of a base pair. The lack of this base pair makes the A-site less susceptible to binding by aminoglycosides, which can be used for the treatment of bacterial infections. Due to the presence of C1494 and A1555, which do not pair, the structure of human 12S mt-rRNA is similar to that of the P^R^ mutant yeast strain 15S mt-rRNA. However, mutations in these positions, namely 1555AàG, which pairs with C1494, or 1494CàU, which pairs with A1555, lead to a stabilized hairpin structure that allows the binding of paromomycin and other aminoglycosides, conferring aminoglycosides sensitivity [133]. These mutations are associated with maternal inherited nonsyndromic deafness or, in most cases, with aminoglycoside-induced deafness [134,135].

Mutations conferring paromomycin resistance in yeast affect the phenotype of mutations in genes encoding for some mitochondrial tRNAs modifying enzymes. As reported in the next two sections, mutations in such genes do not affect the respiratory phenotype of a wild type, paromomycin sensitive (P^S^) strain, but inhibit OXPHOS activity in the mutant, paromomycin resistant strain. Due to the similarity between yeast P^R^ 15S mt-rRNA and human 12S rRNA A-sites, it is possible to exploit yeast to perform complementation analyses of such genes in a paromomycin-resistant background to mimic the normal conditions found in humans (Figure 1) [69].

Different mutations have been studied in P^R^ strains. First, point mutations and deletion in *MSS1* were shown to determine a respiratory impairment, caused by the block in the splicing of introns of *COX1* mRNA due to the inhibition of the synthesis of maturases encoded by mtDNA genes, only in a P^R^ background [27]. Second, deletions in *MTO1* or *MSS1* showed a significant growth defect just in P^R^ strains [24], probably due to the impairment of interaction between the A-site of the ribosome, where the 15S mt-RNA mutation localizes, and the unmodified tRNAs which are the substrate of the Mto1/Mss1 complex [136]. Third, the deletion of *MTO2*, when studied in a P^R^ background, showed an OXPHOS defect due to a defective expression of some mitochondrial genes, among which *COB* and *COX1* [29]. Last, mutations in the *TRM5* gene were studied both in paromomycin-sensitive and paromomycin-resistant backgrounds; in the latter case, a worsening of the phenotype was visible [88], as described in Section 2.3.

### 2.2. MTO1/Mto1, GTPBP3/Mss1 and TRMU/Mto2 Catalyze the 5-Taurinomethylation, or the 5-5- Carboxymethylaminomethylation, and the 2-Thiolation of Uridine at Position 34 of Specific mt-tRNAs

*MTO1* (mitochondrial tRNA translation optimization 1) and *GTPBP3* (GTP binding protein 3) are nuclear genes encoding the enzymes that catalyze the formation of 5-taurinomethyluridine (τm^5^U) (Figure 2) in the first anticodon nucleotide of specific mitochondrial tRNAs, including tRNA^Gln(UUG)^, tRNA^Glu(UUC)^, tRNA^Lys(UUU)^ and, putatively, of tRNA^Leu(UAA)^ and tRNA^Trp(UCA)^ [137,138]. This modification is coupled to the 2-thiolation of U34, forming 5-taurinomethyl-2-thiouridine (τm^5^s^2^U) (Figure 2), performed by the gene product of *TRMU* (or *MTU1* or *MTO2*) in tRNA^Gln(UUG)^, tRNA^Glu(UUC)^ and tRNA^Lys(UUU)^ [28].

Post-transcriptional modifications of U34 increase the fidelity and the efficiency of mtDNA translation through impact on the tRNA structure, interaction with the ribosomal complex, the stabilization of the codon-anticodon pairing, and tRNA recognition by the correct aminoacyl-transferase (summarized in [60]). Human *MTO1* was isolated and characterized starting from the yeast *S. cerevisiae* orthologous gene *MTO1*, whose protein sequence was subjected to a BLAST search to identify human homolog. The predicted human MTO1 protein contains 692 amino acids, a typical mitochondrial target sequence, and a predicted molecular mass of 77 kDa. The protein is well conserved through evolution; in fact, the sequence identity is also higher than 50% between *S. cerevisiae* and *E. coli* [23]. *MTO1* gene maps to chromosome 6q13, and contains 12 exons: five transcriptional variants were identified, resulting from the alternative splicing of exons 4, 6A and 12A or intron 5. Transfection experiments showed that MTO1 localizes exclusively to mitochondria. Northern blot analysis detected a 2.6-kb *MTO1* transcript in all the tissues examined, with the highest expression in tissues with high metabolic rates. An additional transcript was also identified in the liver and skeletal muscle [23].

Five variants of *GTPBP3*, containing a typical mitochondrial targeting pre-sequence, were identified starting from the yeast *MSS1* gene. The variants are due to alternative splicing and the use of different polyadenylation signals. The *GTPBP3* gene maps to chromosome 19p13.11 and contains nine exons. Two variants differ in their 3′-UTRs and encode an identical 492-amino acid protein with a predicted molecular mass of 52 kDa. Two other variants encode an identical 471-amino acid protein, and another variant encodes a longer 524-amino acid protein. [25]. The exons 1, 7, 8 and 9 are those subjected to alternative splicing and the differences in protein length are due to the lengths of G domains [139]. The protein is well conserved through evolution, including bacteria, yeast and nematodes. GTPBP3 localizes exclusively to mitochondria. Northern blot analysis detected the ubiquitous expression of 2.6- and 1.9-kb transcripts, with the larger transcript being more abundant in most tissues [25].

*TRMU* was identified by searching for sequences similar to the yeast *SLM3*/*MTO2* gene (hereafter, *MTO2*). The cDNA ORF spans 1266bp and encodes for a 421 amino acids long protein with a predicted molecular mass of 48 kDa [29]. *TRMU* gene maps on chromosome 22q13 and contains 11 exons. Three *TRMU* splicing variants differing for 99 amino acids were identified; all variants lack exon 3, and one also lacks exon 10, resulting in a truncated protein. Through Northern blot analyses, expression levels were assessed, identifying a 1.9-kb transcript characterized by ubiquitous expression, with the highest levels identified in tissues with high metabolic rates [85].

In *E. coli*, GidA (also known as MnmG) and MnmE are the orthologs of MTO1 and GTPBP3, respectively, and constitute a functional hetero-tetrameric complex and perform the flavin adenine dinucleotide (FAD) and the GTP-dependent incorporation of the carboxymethylaminomethyl (cmnm) group at position five of the first anticodon nucleotide (U34) of tRNA^Lys^, tRNA^Glu^, tRNA^Gln^, tRNA^Leu^, tRNA^Arg^ and tRNA^Gly^ [139,140,141,142]. In tRNA^Lys^, tRNA^Glu^ and tRNA^Gln^, MnmA, the ortholog of *TRMU*, performs the thiolation at position two of U34 [143]. While the modification in position two is important, especially for the aminoacylation of tRNAs, the modifications in both position two and five are essential for the correct codon recognition [144]. GidA structure was resolved in vitro. This highly conserved protein is constituted of three domains: the N-terminal FAD-binding domain, the insertion domain and the α-helical domain. Two disordered loops are also present. In addition, a deep cleft was identified between the FAD-binding domain and the insertion domain. The cleft is characterized by at least seven conserved basic residues, involved in the interaction with the cognate negatively charged tRNA molecule. The C-terminal region is not visible in the crystalized structure, but it presents a key role in the interaction with MnmE [145]. The MnmE structure has also been solved; in particular, this protein is composed of the N-terminal α/β domain, involved in homo-dimerization, a central helical domain and the G domain [146]. The α/β domain constitutes a binding site for tetrahydrofolate (THF), which acts as a one-carbon unit donor in the reaction. The helical domain is poorly conserved, except for the C-terminal part that is involved in the tRNA modification [140]. The G domain is characterized by the classic Ras-like fold and is constituted by the typical G1 to G4 motifs. MnmE is an atypical GTPase; in particular, it shows a peculiarly high intrinsic GTPase hydrolysis rate and a low affinity for GTP and GDP. The GTPase activity was shown to be essential for the functionality of this protein on tRNA modification [139]. Both in *E. coli* and humans, GidA/MTO1, and not MnmE/GTPBP3, seem to be involved in the interaction with the cognate tRNA. However, while in *E. coli* and yeast GidA/Mto1 and MnmE/Mss1 catalyze the 5-carboxymethyl-aminomethylation of uridine, resulting in cmnm^5^U, in humans, MTO1 and GTPBP3 catalyze the 5-taurinomethylation of uridine, resulting in τm^5^U. In bacteria and yeast, 5,10-methylenetetrahydrofolate (5,10-CH_2_-THF) is used as the methyl donor and glycine as the substrate which is added to the methyl moiety. In humans, the τm^5^U modification is synthesized starting from 5,10-CH_2_-THF and taurine instead of glycine. However, in conditions of taurine starvation, the human MTO1/GTPBP3 complex can use glycine as substrate, resulting in cmnm^5^U modification. Potassium, GTP and FAD act as cofactors in all cases [6,28,60].

MnmA presents a cradle-like shape structure consisting of three domains: the N-terminal catalytic domain, the central domain and the C-terminal domain. The catalytic domain is characterized by a mixed α/β fold, contains a P-loop motif and shares similarities with the N-type ATP pyrophosphatases [147,148,149]. MnmA domains are complementary to the L-shaped tRNA, allowing contact with the anticodon arm and the D stem. The interactions between MnmA and the cognate tRNA occur at the level of the minor groove thanks to a Gly-rich region and are stabilized by an Arg243 interaction with the major groove. U34 enters the active site after the recognition by the well-conserved Gln151. Several conserved amino acids also promote the recognition and stabilization of tRNA binding. Subsequently, structure variations occur to assume the pre-reaction and the adenylated conformations. The catalytic mechanism occurs in two steps; first, the C2 atom of U34 is activated through adenylation using ATP; second, the reductive elimination of sulfur from the cysteine by a cysteine desulphurase forms an enzyme-bound cysteine-persulphide intermediate which, in turn, is delivered by a sulfur relay to Cys199 of MnmA. Finally, the thiolic group is transferred to the activated C2 atom of U34, with the support of Cys102 [149], resulting in s^2^U modification. In yeast and humans, the two cysteines are conserved in TRMU/Mto2 proteins, suggesting a similar mechanism.

Mutations in *MTO1* are responsible for combined oxidative phosphorylation deficiency 10 (COXPD10, OMIM #614702), characterized by autosomal recessive inheritance, and resulting in variable defects of the mitochondrial oxidative respiration. The pathology is usually characterized by infancy onset and patients present hypertrophic cardiomyopathy and lactic acidosis. Several patients have been identified as being affected by hypertrophic cardiomyopathy and lactic acidosis due to mutations in the *MTO1* gene. Ghezzi and coauthors [69] reported three patients, including two sibs born from unrelated parents. The first presented lactic acidosis and hypoglycemia after birth and complex III and IV levels were reduced while the second, the younger sister of patient one, presented severe metabolic acidosis and increased blood lactate since birth. Complex I and complex IV activity were severely impaired. The third patient, unrelated to the two sibs, showed a similar disorder but was characterized by a better outcome, with a significant impairment of complex I and IV activity. Five additional patients, from three unrelated families, were reported by Baruffini et al. in 2013; these patients were affected by COXPD10 and presented infantile cardiomyopathy and lactic acidosis. Three patients presented with poor feeding, hypotonia and a failure to thrive; two sibs presented cardiomyopathy. The disease course was highly variable, two patients died and the other three are alive, presenting stable cardiac disease. An impairment in complex I and IV was identified, causing defects in mitochondrial respiration [70]. Another 34 patients were identified by different collaborators and their characteristics are summarized by Zhou and coauthors [74]. In the 42 patients reported, 25 *MTO1* variants were identified: 17 missenses, five frameshifts, two deletions and one duplication. Two variants, Ala428Thr and Thr411IIe, were frequent in the 16 patients who reported death, suggesting that these mutations could lead to more severe impairment of the mitochondrial translation. The main clinical features are lactic acidosis, cardiovascular, combined complex deficiency, developmental delay, hypotonia, intellectual disability, delayed psychomotor development and abdominal disease/feeding difficulties. Encephalopathy, optic atrophy, ataxia, cognitive impairment, poor speech and spasticity are also present in a fewer number of patients [74]. Complex I and IV deficiencies were the most common of the combined deficiencies [72].

Mutations in *GTPBP3* are responsible for combined oxidative phosphorylation deficiency 23 (COXPD23, OMIM #616198), characterized by autosomal recessive inheritance. The pathology is usually characterized by the early childhood onset of hypertrophic cardiomyopathy and/or neurologic symptoms, including delayed psychomotor development and hypotonia [77]. Fifteen patients carrying mutations in *GTPBP3* were reported through the years. Kopajtich and coauthors reported 11 patients from nine families affected by combined oxidative phosphorylation deficiency [77]. All patients presented lactic acidosis, nine presented cardiomyopathy and six had variable neurologic symptoms, including feeding difficulties in infancy, hypotonia, developmental delay and intellectual impairment. Muscle biopsies from most patients showed severe deficiencies in complexes I and IV activity. Mutations in eight patients were discovered by whole-exome sequencing of a cohort of 790 individuals with suspected mitochondrial disease. One patient was identified through whole exome sequencing, presenting a mild form of COXPD23 [150]. Three other patients were identified through whole genome sequencing and targeted panels of candidate human mitochondrial genes [78]. All these patients can be classified into two categories; patients presenting a severe form of COXPD23, who died in the first nine months of life, and patients presenting a mild form of COXPD23, who may survive the second decade of life. Six mutations were identified in the severe form, causing the loss of TrmE-type G domain, or substitutions in TrmE-type G domain or at the N-terminus of GTBBP3 protein. Seven mutants were identified in the mild form, while a mutation was observed in both severe and mild forms.

Mutations in *TRMU* are related to two distinct groups of pathologies. Homozygous and compound heterozygous variants are the cause of transient infantile liver failure (LFIT, OMIM #613070). Mutations in this gene impair mitochondrial protein synthesis and oxidative phosphorylation, determining the disease. A total of sixty-two patients carrying mutations in *TRMU* have been reported. Zeharia and coauthors first described 13 individuals with mutations in this gene; four of them died and the others showed no further neurologic or hepatological issues over the years of follow-up [79]. Subsequently, another 49 cases were as summarized in [84]. Forty-eight different variants were identified, and the catalytic domain was the most affected; Val259Met and Tyr77His, carried by 15 and 13 patients, respectively, were the most frequent missense variants. The most frequent feature was liver involvement and other frequent features were lactic acidosis, abnormal body weight, emesis and/or diarrhea, nervous system involvement, muscular hypotonia and abnormal growth. Multiple organ failure or hepatic failure were the most frequent causes of death. Considerably reduced respiratory complexes I, III and IV activity was identified in patients affected by LFIT [79]. Furthermore, *TRMU* acts as a nuclear modifier to modulate the clinical manifestation of deafness-associated 12S mt-rRNA mutations (OMIM #580000); aminoglycoside-induced deafness is a maternally inherited pathology that determines sensorineural deafness, ranging from hearing loss to moderate hearing impairment, with no myopathy or neurological symptoms [151,152]. The 12S mt-rRNA was identified as the main gene that, if mutated, determines the disease; in particular, A1555G mutation determines a conformational change that results in increased aminoglycoside binding, causing a greater susceptibility to the effects of these antibiotics on the translational fidelity [134] and resulting in aminoglycoside-induced ototoxicity and deafness. *TRMU* was identified as a nuclear modifier gene for the phenotypic expression of 12S mt-rRNA-related deafness; in particular, *TRMU* carrying the mutation p.Ala10Ser, leading to a mitochondrial translation impairment, and the A1555G mutation in 12S mt-rRNA modulates the manifestation of the pathology [85]. Impaired U34 modification due to hypomorphic *TRMU* alleles worsens the translation impairment caused by the A1555G variant, determining impaired respiratory activity and, subsequently, elevated ROS production and reduced ATP synthesis [86].

Yeast Mto1 and Mss1 are responsible for the biogenesis of the 5-carboxymethylaminomethyl (cmnm^5^U34) group (Figure 2) of the wobble uridine base of mt-tRNA^Lys(UUU)^, mt-tRNA^Glu(UUC)^ and mt-tRNA^Gln(UUG)^, and the impairment of Mss1 and Mto1 activity can determine the loss of this modification [28,60]. Mto2, the ortholog in yeast of the human TRMU, is responsible for the 2-thiolation of the wobble uridine base, resulting, together with the modification catalyzed by Mto1/Mss1, in the mnm^5^s^2^U34 (Figure 2) of mitochondrial tRNA^Lys(UUU)^, tRNA^Glu(UUC),^ and tRNA^Gln(UUG)^ [28]. The deletion of *MTO2* determines the instability of mt-tRNAs and the subsequent impairment in the aminoacylation of the tRNAs [153]. The deletion of *MTO2* and of *MTO1* or *MSS1*, but not of the single gene, results in the inability of growing on oxidative carbon sources, as a consequence of reduced levels of the three mt-tRNA substrates of the enzymes, low levels of *COX1* and *COB* mRNA and the complete loss of mitochondrial protein synthesis.

The P^R^ yeast strain harboring mutation C1477G in the 15S mt-rRNA showed resistance to paromomycin but hypersensitivity to neomycin, another aminoglycoside; in fact, transcription levels of mtDNA, respiratory activity and membrane potential were severely decreased after neomycin treatment. The *mto2* null mutant strain reduces hypersensitivity to this antibiotic, suggesting an interaction between the 2-thiolic group introduced by Mto2 and the A-site of 15S mt-rRNA. These findings show the involvement of *MTO2* in the modulation of aminoglycoside sensitivity in the P^R^ background in *S. cerevisiae*, as in humans harboring mutations in the corresponding region of the 12S mt-tRNA [154].

Thanks to the presence of *MTO1* in yeast, it was possible to exploit this model organism to study the alleged pathological mutations. Human mutations p.Arg620Lysfs*8 and p.Ala428Thr were studied using the strains carrying the respective mutations in yeast *MTO2*, P622* and A431T [69]. In a genetic background containing a wild type 12S rRNA (P^S^ strain), the *mto1* null mutant strain showed a slight decrease in the oxygen consumption rate, and expression of the *mto1^P622*^* allele does not recover this phenotype, indicating that the variant truncated at amino acid 620 is functionally inactive. The decreased oxygen consumption rate was only partially rescued by the *mto1^A431T^* allele compared with the full rescue by the *MTO1* wild type allele, also showing that this mutation determines pathological defects. However, no oxidative growth defect was visible for all the strains. To better understand the role of these mutations in human disease, mutations under analysis were studied also in the P^R^ background, mimicking the wild type human condition of the small subunit mt-rRNA, as reported in Section 2.1. In this case, the *mto1* null strain was not able to grow on oxidative carbon sources such as glycerol. No oxidative growth was visible for the P622* mutant, while a leaky growth defect was visible for the strain carrying the A431T variant compared with the fully restored growth obtained with the expression of the wild type allele. The respiratory activity results parallelized oxidative growth analyses, supporting the pathogenicity of these mutations. To confirm the pathogenicity of a second group of variants identified in five patients, P^R^ *mto1* null strain carrying *mto1^T414I^* and *mto1^R481H^* mutant alleles, equivalent to human mutations p.Thr411Ile and p.Arg477His, respectively, were studied [70]. No oxidative growth defect was detectable for the strain carrying the R481H variant, while a severe growth defect was determined by the T414I variant. Evaluating the respiratory activity, a severe impairment was identified for T414I variant, similar to that of the null mutant, and a leaky oxygen consumption rate reduction was detected for R481H variant, confirming the pathogenicity of both mutations.

Structural analysis showed that substitution with isoleucine of Thr414 may alter the position of serine 371, involved in the catalytic process through binding or stabilization of FAD; therefore, in an attempt to rescue the phenotype with a small molecule, the effect of riboflavin supplementation was evaluated on the T414I mutant. No effect was visible on oxidative growth and respiratory activity, suggesting that the catalytic activity of the *mto1^T414I^* gene product is fully impaired [70].

Molecular consequences were assessed by performing mitochondrial protein synthesis in the P^R^ background. No defects in the de novo mitochondrial protein synthesis profile were observed in the strain harboring the A431T or R481H variants. On the contrary, the strain without *mto1* or harboring the P622* or the T414I variants showed normal levels of newly synthetized Var1, Cox3, Atp6 and Atp8/9, whereas Cox1, Cox2 and Cob were absent, indicating that complex III and IV should be impaired. Indeed, complex IV activity was abolished in the strains with P622* and T414I variants, whereas a significant reduction of the activity was observed for the strain carrying the A431T variant compared with the strain carrying the *MTO1* wild type allele, and no impairment in CIV activity was observed for the strain carrying the R481H variant. Functional analyses in yeast allowed the validation of the human variants identified in patients and the severity of the phenotype are in accordance with the clinical manifestation in patients, supporting the utility of yeast as a model for the validation of alleged pathological mutations. In agreement with this hypothesis, the most deleterious mutations are associated with a strong defect in protein synthesis, whereas the mild mutations are associated with normal levels of protein synthesis; the latter observation leads to the hypothesis that the mitochondrial protein are still synthesized but their sequence may be altered due to incorporation of the wrong amino acids due to defect in the codon–anticodon pairing when the cmnm^5^ modification does not occur [70].

To date, no mutations in *MSS1* and *TRMU* allegedly associated with primary mitochondrial diseases have been modeled in yeast, although yeast has been used, as reported previously, for studying a possible effect of mutations in the genes reported in this section with aminoglycoside-induced ototoxicity and deafness.

### 2.3. TRMT5/Trm5 Catalyzes the 1-Methylation of Guanosine at Position 37 of Specific mt-tRNAs

*TRMT5*, also known as *TRM5*, is a nuclear gene encoding the tRNA methyltransferase 5. *TRMT5* methylates the N1 position of guanosine 37 (m^1^G) (Figure 2), the nucleoside adjacent to the anticodon, in several tRNAs that present such nucleotide in position 37, including the well-characterized cytoplasmic tRNA^Leu(GAC)^ and the mitochondrial tRNA^Pro(UGG)^ using S-adenosyl methionine (AdoMet) as substrate [34]. The methylation of G37 plays a crucial role in preventing frameshift errors at the ribosome level during elongation [155,156]. *TRMT5* was first discovered by sequencing clones deriving from a size-fractionated cDNA library from the human brain and called KIAA1393 [157]. Through a BLAST search using yeast *TRM5* sequence, KIAA1393 was found as a homologous human sequence, encoding a protein made up of 500 amino acids, and was renamed *TRMT5*. *TRMT5*, which maps to chromosome 14q23.1, contains five exons with an additional upstream exon with an importation signal that could enable the protein localization in the mitochondria, to allow the methylation of mitochondrial tRNAs [34]. TrmD, the enzyme that performs the methylation in position N^1^G37 in bacteria, has a size of 28 kDa and functions as a homodimer [158]: the interacting surface between the two subunits composes the catalytic center. Archaeal and eukaryotic tRNA methyltransferases were also identified; the archaeal Trm5s are 29–39 kDa, while the eukaryotic Trm5s are 56–58 kDa, due to the C-terminal and N-terminal extensions. The overall similarity between the archaeal and eukaryotic tRNA methyltransferases reaches 60%. Archaeal and eukaryotic Trm5s belong to the class-I methyltransferases and are likely to have the Rossmann fold structure for catalytic activity [159,160]. Archaeal and eukaryotic Trm5s and bacterial TrmD appear to be evolutionally unrelated to each other. Different ways of tRNA recognition between archaeal and eukaryotic Trm5s and TrmD have been revealed, supporting the proposal that Trm5s and TrmD have evolved from different ancestral enzymes [159,161,162]. For example, G36 residue is necessary to have the correct methylation of G37 in bacteria [163], while Trm5s methylate G37 whatever the previous base was [35,164]. The archaeal Trm5 structure has been resolved and consists of three domains, called D1, D2 and D3 through the crystallization of the *Methanocaldococcus jannaschii* protein. D3, the C-terminal domain, defining the class I methyltransferases, is characterized by the Rossmann fold and binds the substrate AdoMet. D2 is the central domain associated with D3, and these two domains play the leading role in tRNA methylation. Moreover, D1, the N-terminal domain, not conserved through evolution, is structurally independent of D2-D3 [165]. This enzyme catalyzes the reaction whereby guanosine 37 of the tRNA and S-adenosyl-L-methionine lead to the production of N1-methylguanosine 37 and S-adenosyl-L-homocysteine, with the release of H^+^.

Mutations in this gene are responsible for peripheral neuropathy with variable spasticity, exercise intolerance and developmental delay (PNSED, OMIM #616539), characterized by autosomal recessive inheritance. This pathology is a multisystemic disorder with variable manifestations, within the same family too. Some patients show symptoms with infantile/childhood onset. Among the symptoms, there are hypotonia and global developmental delay with poor or absent motor skill acquisition and poor growth. Other patients present symptoms as young adults with exercise intolerance and muscle weakness. All patients have peripheral neuropathy, usually demyelinating, with distal muscle weakness, atrophy and distal sensory impairment. Spasticity, extensor plantar responses, contractures, cerebellar signs, seizures, short stature and the rare involvement of other organ systems are additional features of this pathology. Deficiencies in mitochondrial respiratory complex enzyme activities in patient tissues are sometimes identified through biochemical analyses, despite the fact that they are not always apparent. As a mitochondrial dysfunction hallmark, lactate is frequently increased [88,90].

Eight patients have been identified with PNSED; two unrelated patients with a high variable phenotype, caused by a defective mitochondrial respiratory chain activity, were reported [88]. One patient, a female, previously reported by Haller and coauthors [166], showed lactic acidosis and a mitochondrial myopathy associated with a deficiency of complexes III and IV. The other patient, a 7-year-old male showed the symptoms since birth. Among various symptoms, he showed decreased activity of mitochondrial complex IV, and low complex I activity [88]. Then, two sisters with variable neuromuscular abnormalities with childhood-onset were reported [89]. Mitochondrial respiratory chain analyses showed complex I, III and IV deficiencies with a more severe phenotype in the younger sister. Both patients showed intermittently increased lactate levels. In 2022, four new patients were reported; three unrelated patients presented the pathology during infancy or early childhood with global developmental delay. Mitochondrial respiratory chain analyses showed mild complex I deficiency only in a single patient [90]. The last patient, a female from a Chinese family showed, in addition to a severe developmental delay, several symptoms, different from those identified in the other patients, further expanding the clinical spectrum of this pathology. For the first seven patients, the mutations were identified through whole genome sequencing (WES), whereas for the last patient medical exome sequencing was performed [167].

*TRM5*, the yeast ortholog of human *TRMT5*, encodes for a protein that is responsible for the methylation of both cytoplasmatic and mitochondrial tRNA. In cytoplasm, Trm5 methylates cytoplasmic tRNA^Leu(UAR)^, tRNA^Pro(NGG)^, tRNA^Arg(CGG)^, tRNA^His(GUG)^, tRNA^Asp(GUC)^ [35] and tRNA^Ala(IGC)^ that presents, after modification catalyzed by the adenosine deaminase Tad1 [168], inosine in position 37 (m^1^I37). In addition, it participates, catalyzing the methylation at the N1 position, to the formation at position G37 of tRNA^Phe(GAA)^ of the deeply modified nucleoside wybutosine, an unusual nucleoside deriving from guanosine containing three rings, whose aim it is to restrict the flexibility of the anticodon and whose synthesis requires at least five genes [169,170,171,172]. In mitochondria, Trm5 methylates the initiator tRNA^fMet(CAU)^ and tRNA^Phe(GAA)^ [173], and putatively tRNA^His(GUG)^, tRNA^Leu(UAA)^, tRNA^Phe(GAA)^, tRNA^Pro(UGG)^, tRNA^Ser(UGA)^ and tRNA^Thr(UAG)^ [174].

Thanks to the presence of the human ortholog in yeast, the pathogenicity of two variants, p.Arg291His and p.Met386Val, found in different patients in compound heterozygosis with the frameshift mutation p.Ile105Serfs∗4, was confirmed [88]. Both patients presented with a decreased activity of the respiratory complexes in muscles cell and decreased methylation of G37 of mt-tRNA^Leu(UAG)^. With regard to the structural aspect, Arg291 amino acid is in close proximity to Glu288, a conserved amino acid involved in the catalytic domain of the protein, so that a substitution of this could affect the interaction with residue 288, leading to a destabilization of the active site. Met386 does not have a direct catalytic role, but a small aliphatic amino acid, such as valine, may alter the position of Asn387, an amino acid of the catalytic pocket. Although the structural analysis suggested a pathogenic role of these mutations, an ad hoc functional analysis was necessary to correlate the role of these mutations with the defects in the respiratory complexes observed in patients. In the study, yeast was exploited to validate the pathogenicity of these two mutations, constructing a strain expressing mitochondrial *TRM5* wild type or mutant alleles harboring the equivalent mutation, R270H and M396V, respectively. Since Trm5 acts both on cytoplasmatic and mitochondrial tRNAs, the *trm5* null mutant is lethal. To bypass the lethality and study the effect only on mt-tRNAs, a strain lacking the gene fragment which encodes for mitochondrial targeting sequence was created (*trm5Δ1-33*). This strain showed only a leaky respiratory deficiency, suggesting that methylation of 37G in mt-tRNAs is not critical in yeast for proper mitochondrial protein synthesis. However, as in the case of *MTO1*/*MSS1* and *MTO2*, the respiratory deficiency was exacerbated in the P^R^ strain described in Section 2.1, suggesting that the modified base at position 37 may interact with the region of the A-site of 15S mt-rRNA conferring resistance or susceptibility to aminoglycosides. The respiratory activity was rescued by the expression of wild type *TRM5*, whereas strains carrying the mutations R270H or M396V led just to a partial rescue of the respiratory activity, suggesting that they are the cause of the pathogenicity in patients when in compound with a null allele.

### 2.4. TRIT1/Mod5 Catalyzes the N6-Isopentenylation of Adenosine at Position 37 of Specific mt-tRNAs

*TRIT1* encodes for the human tRNA isopentenyl transferase (IPTase), an enzyme of 50 kDa responsible for the addition of an isopentenyl group at position N6 of adenosine 37 (i^6^A) (Figure 2), in several cytosolic and mitochondrial tRNAs that present such nucleotide in position 37. The gene was identified by searching in an EST database sequences similar to that of the *S. cerevisiae MOD5*, the yeast orthologous of *TRIT1*, followed by amplification by PCR and 5-prime RACE from a human kidney cDNA library [40]. *TRIT1*, which maps to chromosome 1p34.2, contains 11 exons, which undergo alternative splicing, resulting in at least five isoforms. Isoforms 1 and 4 are targeted to mitochondrion, whereas isoforms 2, 3 and 5 are cytoplasmatic. The main isoform is 467-amino acids long and contains four conserved isopentenyl transferase motifs, including an ATP/GRP motif A and a putative dimethylallylpyrophosphate-binding site. It is also characterized by a bipartite nuclear localization signal and a C-terminal C2H2-type zinc finger motif. In particular, the human tRNAs modified by TRIT1 are the cytosolic tRNA^Ser(AGA)^, tRNA^Ser(CGA)^, tRNA^Ser(UGA)^ and tRNA^Sec(UCA)^, which inserts selenocysteine in specific proteins, and the mitochondrial tRNA^Cys(GCA)^, tRNA^Trp(UCA)^, tRNA^Tyr(GUA)^, tRNA^Ser(UGA)^ and tRNA^Phe(GAA)^ [175]. All the tRNAs have a stretch of three adjacent A (A36, A37 and A38), a condition that is necessary but not sufficient for the modification at position 37. The modification increases tRNA affinity to the ribosome, can activate mRNA decoding and decrease frameshifting and stabilizes the weak codon-anticodon pairing at the adjacent position A36 [176,177].

The homologous tRNA isopentenyl transferases have been well conserved during the evolution from bacteria to humans; however, several changes have occurred in order to also target the proteins to mitochondria. Biochemical studies and the structures of bacterial and *S. cerevisiae* IPTases bound to their substrates have shown a direct recognition of the anticodon loops (ACLs) of their target tRNAs, and the i^6^A37 formation using dimethylallyl-pyrophosphate (DMAPP) as an isopentenyl donor. The eukaryotic dimethylallyl-pyrophosphate transferase structure is composed of a core domain, remarkably similar to that of the bacterial enzyme, and of an insertion domain with a five-helix bundle and connected to the core domain through two loops. It is also composed of the C-terminal extension that can be divided into two parts and this is unique to the eukaryotic enzyme. The C-terminus is a zinc finger connected to the core domain trough a linker region of 30 amino acids and composed of two short helices and loops traversing along the core domain. Interaction between tRNA and DMAPP transferase is extensive and is mainly concentrated at the anticodon stem-loop [178,179,180,181,182].

Mutations in this gene lead to a pathology called combined oxidative phosphorylation deficiency 35 (COXPD35, OMIM #617873), a rare autosomal recessive disease caused by homozygous or compound heterozygous mutations in the tRNA isopentenyl transferase gene *TRIT1* on chromosome 1p34. To date, only eight cases and seven allelic variants have been reported of this disorder. The affected patients are characterized by variable deficiencies in mitochondrial respiratory enzyme complexes and the most frequent symptoms observed are intellectual disability, hypotonia, microcephaly, global development delay and central nervous system involvement. These symptoms can occur in early infancy and the severity can be variable. The mutations identified in these patients are p.Arg323Gln, p.His419Pro, p.Ile283Ser, p.Arg402Ter, p.Lys286Glu, p.Arg8Ter and p.Glu327Lys. The p.Arg323Gln mutation was identified by whole-exome sequencing in two siblings born from consanguineous parents of Pakistani origin [41]. The p.His419Pro and p.Ile283Ser mutations were identified in a 16-year-old girl in compound heterozygous. The p.His419Pro mutation was also found in an unrelated patient in compound with the p.Arg402Ter mutation, and both mutations were found by WES (Kernohan et al., 2017). The p.Lys286Glu missense mutation was found in two siblings with compound heterozygous mutations p.Lys286Glu and p.Arg8Ter and these mutations were also found by WES [91]. The p.Glu327Lys mutation was found in two Korean siblings by WES and confirmed by Sanger sequencing [183].

*MOD5* is the yeast homolog of the human *TRIT1* and the similarity between the two proteins is 53%. Besides its enzymatic activity, the yeast counterpart furthermore binds the RNA polymerase III transcription complexes on nuclear tRNA genes helping the “silence” of the transcription of RNA polymerase II [184]. Unlike human IPTase, the yeast Mod5 modifies five cytosolic tRNAs (tRNA^Ser(AGA)^, tRNA^Ser(CGA)^ and tRNA^Ser(UGA)^, as in humans, and the specific tRNA^Tyr(GUA)^ and tRNA^Cys(GCA)^), and three mitochondrial tRNAs (tRNA^Trp(UCA)^, tRNA^Tyr(GUA)^ and tRNA^Cys(GCA)^) instead of five as found in humans [185]. In the fission yeast *Schizosaccharomyces pombe*, it was shown that i^6^A37 increases the translational efficiency and the fidelity in a codon-specific manner cognate with tRNAs containing the i^6^A37 [185].

Using the yeast model has been critical to confirm the pathogenicity of a homozygous variant harboring mutation p.Arg323Gln found in two siblings with encephalopathy and myoclonic epilepsy due to multiple OXPHOS deficiencies in skeletal muscle, born from consanguineous parents of Pakistani origin as reported above. Studies on patients’ cells showed a decrease in the respiratory complex activity, in the in vivo mitochondrial protein synthesis, in the steady state levels of several subunits of respiratory complexes, and in the levels of cytosolic and mitochondrial tRNA harboring i^6^A37 modification. To evaluate whether these defects correlated with a defective oxidative phenotype, an *S. cerevisiae* strain deleted in *MOD5* was generated and transformed with a *mod5* allele harboring arginine instead of the naturally occurring lysine at position 294, equivalent to p.Arg323 in TRIT1 (humanized *MOD5*) or a *mod5* allele harboring glutamine at position 294 as found in patients (*mod5^K294Q^* mutant allele). The yeast strain transformed with the mutant allele showed reduced growth in a medium supplemented with ethanol, a non-fermentative carbon source, compared with the strain with the humanized *MOD5*. The respiratory activity was reduced in the mutant strain compared with the humanized strain as well, indicating that the growth defect on oxidative carbon sources was due to impaired respiratory activity. The correlation between enzymatic defect found in patients’ cells and a respiratory phenotype was confirmed also in another yeast model, the fission yeast *Schizosaccharomyces pombe* deleted in the orthologous gene *TIT1* and transformed through heterologous complementation with a human *TRIT1* wild type and the mutant allele (Yarham et al. 2014). The wild type allele, but not the mutant one, rescued the oxidative growth defect on a medium supplemented with the non-fermentable carbon source glycerol. In addition, a defect in the enzymatic activity was confirmed by the red–white assay in *S. pombe*. The red-colored phenotype in the presence of low adenine in the medium was due to nonsense mutation in *ADE6*. The *ADE6* mutant allele contains a premature UGA stop codon, which can be suppressed by the suppressor tRNA^Ser(UCA)^, an isoform derived from the gene for the tRNA^Ser(UGA)^ after a G→C transversion provided that it, as does its human counterpart, contains i^6^A37, which is critical for the pairing of the anticodon to UGA and thus for the translational activity of the suppressor-tRNA [186]. Colonies harboring the mutant variant were red in presence of low levels of adenine in the medium, indicating that no suppression occurs due to the lack of isopentenylation, whereas colonies harboring the wild type variant were of their naturally occurring color, white [41].

### 2.5. TRNT1/Cca1 Catalyzes the CCA Trinucleotide Addition at the 3′ of All the mt-tRNAs

*TRNT1* encodes for the tRNA-nucleotidyltransferase, an enzyme involved in the addition of the CCA trinucleotide to the 3′ end of all mitochondrial and cytoplasmic tRNAs. Although, strictly speaking, the addition of the CCA stretch is not a nucleotide modification, here we discuss also the effects of *TRNT1* mutations since the addition of CCA is considered an ad hoc modification that occurs after transcription and processing of the pre-tRNAs [61]. Indeed, this modification is required for the aminoacylations of tRNAs and for the correct positioning of the peptidyl-tRNA at the P-site and the aminoacyl-tRNA at the A-site on the large ribosomal subunit [187,188]. In addition, CCA-adding enzymes are also involved in the tRNA quality control and stress response [189,190,191,192].

*TRNT1*, which maps to chromosome 3p26.2, contains seven exons and encodes for at least one cytoplasmic isoform and two mitochondrial isoforms. *TRNT1* is an essential gene: siRNA knockdown of *TRNT1* in human fibroblasts is responsible for cytotoxicity and apoptosis [54], while the deletion of the yeast ortholog, called *CCA1*, is lethal. Although the catalyzed reaction is identical, there are important differences found in the three kingdoms of life both in the overall structure and in the organization of the catalytic core [92,193]. These enzymes can be classified into two classes of the polymerase β superfamily: class 1, found in archaea, and class 2, found in bacteria and eukaryotes. The crystal structure of the enzyme belonging to class 1 was obtained thanks to the isolation from *Archaeoglobus fulgidus*, a 51 kDa protein containing 437 amino acids and its nucleotide complexes with ATP, CTP and UTP [194]. The structure is composed of four domains named head, neck, body and tail. The head domain has five stranded β sheets flanked by α helices, while the neck domain contains mixed of β sheets and α helices. The body domain is composed of four stranded β sheets flanked by α helices and is connected by a flexible loop to the neck domain that allows a hydrophobic interaction with the neck. The conformation of the tail is stabilized by dimer interactions. Despite the comparable size and the number of domains between enzymes of the two classes, the structural homology concerns the head domain only. The neck, body and tail domains are composed of numerous β strands, contrary to the exclusively α-helical domains found in class 2. In addition, the enzymes belonging to class 1 dimerize via their body and tail domains, while the enzymes belonging to the second class dimerize through a smaller interface formed by their head domain. The tRNA substrate is bound in the cleft between the head and neck domains. In particular, the tRNA acceptor stem lies in the cleft of the two domains, with the 3′-terminus of the tRNA positioned close to the helices responsible for the NTP-binding. In the absence of a bound substrate, the NTP-binding pocket of the enzyme is unable to recognize CTP or ATP specifically. In the class 1 enzymes, after the binding of the tRNA, a conformational switch towards the CTP-binding site occurs and tRNA is elongated by one residue of C with a mechanism that is still unknown. The translocation and rotation enable the binding of another CTP and the addition of the second C-nucleotide, followed by the switching of the NTP pocket to the ATP-binding site. Finally, an ATP binds to the NTP pocket, the A-nucleotide is incorporated and the mature tRNA is released from the complex [195].

Mutations in the *TRNT1* gene cause a pathology called congenital sideroblastic anemias (CSAs, OMIM #616084), which include a group of autosomal recessive syndromic and nonsyndromic inherited diseases [196,197]. These pathologies are characterized by iron deposition in the mitochondria of red blood cell precursors in the bone marrow. To date, 17 pathogenic variants have been reported that lead to a phenotype characterized by retinitis pigmentosa (RP), erythrocytic microcytosis and sideroblastic anemia with B-cell immunodeficiency, development delay and periodic fevers [54,198]. The first mutations, responsible for the first phenotype, were found in consanguineous families by linkage analysis and candidate gene sequencing while the other mutations were found by whole-exome sequencing or direct sequencing of the *TRNT1* gene. The main mutations include substitutions p.Thr154Ile, p.Leu166Ser, p.Arg190Ile, p.Ile233Thr and p.Glu43 in-frame deletion, and several intron mutations which result in altered splicing or frameshift mutations which result in a truncated protein.

In the yeast *Saccharomyces cerevisiae*, a single gene *CCA1* encodes for tRNA CCA nucleotidyltransferase. Aebi and coauthors demonstrated its role after the isolation of a temperature-sensitive (ts) allele, ts352 [199]. Cells bearing this mutation, when shifted to a non-permissive temperature (37 °C), presented an abrupt cessation of protein synthesis due to a reduction in the pool of functional tRNAs [200]. Using multiple transcriptional and translational start sites, nuclear, cytoplasmic and mitochondrial isoforms are produced [201].

The yeast *Saccharomyces cerevisiae* was exploited by examining at first the ability of human *TRNT1* variants, cloned under the *ADH1* promoter, to complement the lethality at high temperatures of the mutant strain harboring the ts352 mutant allele. At first, it was observed that the heterologous expression of the wild type human *TRNT1* allele restored the growth at the non-permissive temperature in a similar way as the wild type yeast *CCA1* allele, indicating that the human enzyme is also capable of modifying yeast tRNAs. *TRNT1* mutant alleles harboring the mutations p.Thr154Ile, p.Met158Val, p.Leu166Ser, p.Arg190Ile, p.Ile223Thr or p.Ile326Val, found in different patients, are able to partially complement the temperature-sensitive growth defect, but at a lower extent compared with wild type *CCA1* or *TRNT1* and, at a different extent, among them, indicating that all the mutations affect the modification of tRNAs. Comparable results were reported by expressing wild type and mutant *TRNT1* alleles, under the *ADH1* promoter or the native *CCA1* promoter, in a strain deleted in *CCA1*. Transformation through plasmid shuffling showed that the wild type *TRNT1* allele restored the viability and the full growth of the strain. The same results were observed for *TRNT1* variants harboring p.Ile326Val and p.Lys416Glu mutations, which are located far from the active site, suggesting that these mutant alleles are milder. On the contrary, *TRNT1* variants harboring the remaining mutations, all close to the active site, restored just partially and at different extents the growth, being the mutations p.Leu166Ser, p.Arg190Ile and p.Ile223Thr the most deleterious. In addition, the latter two mutant strains showed an extremely poor growth on a medium supplemented with a non-fermentable carbon source such as glycerol, suggesting a defect in oxidative phosphorylation likely due to an impairment of the mitochondrial translation. Moreover, the protein levels of all *TRNT1* variants were decreased compared to wild type *TRNT1*, though at different extents and with the exception of the p.Ile326Val variant, whose levels seem to be increased, indicating that the mutation affects the stability of *TRNT1*, as found also in patients’ fibroblasts [54,93].

### 2.6. MRM2/Mrm2 Catalyzes the 2′-O-Methylation of Uridine at a Specific Position of Human 16S mt-rRNA and of Yeast 21S mt-rRNA

*MRM2* (also known as *FTSJ2*) encodes for the mitochondrial rRNA methyltransferase 2, an enzyme of 27 kDa responsible for the methylation of the 2′-O-ribose of U1369 of the 16S mt-rRNA A-loop, resulting in Um (Figure 2), which is essential for the peptidyl transferase activity [202]. The 2′-O-ribose methylation and pseudouridylation are the most common modifications in rRNAs of eukaryotes and archaea. The 16S mt-rRNA includes the 2′-O-methylribose moiety in three different sites, at G1145, U1369 and G1370, and pseudouridylation at U1397. *MRM1* is responsible for the methylation of the first G1145; *RNMTL1* or *MRM3* is responsible for the methylation of G1370; and *RPUSD4* is responsible for the pseudouridylation of U1397 [45,110,203]. These modifications are located in sites that are conserved from bacteria to fungi and animals and contribute to the structure and the activity of the catalytic domain of the mitoribosome, especially the peptidyl transferase activity [107]. All the enzymes modifying the mt-rRNA of the large subunit are conserved in fungi and in most animals, with the exception of *MRM3*, so that the nucleotide equivalent to G1370 is not methylated. For such reasons, the presence of the 2′-O-methylribose moiety in mammalians is unusual, though the corresponding G nucleotide of the yeast cytoplasmatic 23S rRNA is methylated. A possible reason could be that a significant fraction of the mammalian 16S mt-rRNA molecules is not methylated; thus, it is possible that a modification of just a single base, U1369 or G1370, may be sufficient to support mitoribosome activity and translation [107].

*MRM2*, which maps to chromosome 7p22.3, contains three exons, which encode for a single mitochondrial isoform of 27 KDa [204]. *MRM2* belongs to a large family of evolutionarily conserved S-adenosylmethionine-binding proteins, which includes more than 180 members [202]. The human MRM2 shares 34% identity and 52% similarity with the FtsJ/RrmJ of *E. coli*. This enzyme possesses a nucleic acid binding groove, which contains many charged positively amino acids conserved in all species, as well as a conserved S-adenosylmethionine (AdoMet) binding domain. It was also observed that *MRM2* is essential for the biogenesis of the large subunit of the mitochondrial ribosome (mtLSU) due to its capacity to remodel the conformation of 16S rRNA in the late stages of the process: this activity is independent from the methyltransferase activity, since mutations which abolish the latter did not affect the biogenesis [205]. For this reason, a severe mitochondrial dysfunction with defects in mitochondrial translation was observed in the absence of *MRM2*. These defects are caused by the presence of mtLSUs trapped in immature assembly states with unstructured IV and V domains of the mt-rRNA which prevent the initial engagement of the immature particles in translation [205]. Although little is known about the catalytic process of the 2′-O-ribose methyltransferase, a possible mechanism has been proposed for the vaccinia mRNA 2′-O-methyltransferase VP39, that methylates the first transcribed nucleotide in mRNA following the 5′ m^7^Gppp-cap [206]. From the structural analysis of the RrmJ-AdoMet complex, using the mRNA substrate of VP39, it was observed that the ssRNA fits very well into the putative substrate binding site of RrmJ. The most conserved amino acids among the RrmJ homologs are Lys38, Asp124, Lys164 and Glu199, and by comparing the structures of Rrmj with VP39, it results that these amino acids are located at the identical position of VP39, coordinating the phosphate atoms on each side of the methylated nucleotide and probably playing a critical role in the catalysis of RrmJ, forming a catalytic tetrad in the active site. Furthermore, Tyr201, highly conserved among the homologs of RrmJ, is located underneath the putative active site and could play a significant role in the catalytic activity. A possible reaction activity could be based on the deprotonation of the 2′-OH group of the ribose by at least one of two conserved lysine amino acids reported above, followed by a nucleophilic attack of the oxygen to the methyl group of AdoMet. A Mg^2+^ ion seems to be important for the substrate binding and the placement of the 2′-hydroxyl group of the ribose near to the methyl group of AdoMet. 2′-O-ribose methylation on the RNA changes the local properties of the RNA, which can be easily detected [207,208,209,210]. For example, it is possible to measure the resistance to the degradation by RNase H if the RNA is hybridized to a chimeric oligonucleotide, since 2′-O-methylation inhibits the degradation, or to measure the reduction of the PCR products levels after retrotranscription, which is blocked by 2′-O-methylation [107,207].

Mutations in this gene lead to a pathology called mitochondrial DNA depletion syndrome 17 (OMIM #606906), an autosomal recessive pathology characterized by development delay, generalized dyskinesia and ballismus, also involving cervical and oropharyngeal muscles. To date, only one mutation, the p.Gly189Arg, has been reported in homozygosis in a boy with a MELAS-like mitochondrial depletion syndrome, and is absent in the ExAC database. Glu189 is located near the AdoMet binding site and the catalytic site [116].

In the yeast *S. cerevisiae* there are three proteins similar to the *E.coli* 2-O-methyltransferase FtsJ/RrmJ [211]: Spb1, a nucleolar AdoMet binding protein involved in the synthesis of cytoplasmic LSU [212]; Trm7, a cytoplasmatic methyltransferase involved in the anti-codon loop tRNA methylation [213]; and Mrm2, the mitochondrial 2′-O- ribose methyltransferase, which shares 29% identity and 43% similarity with the human *MRM2* [108]. The deletion of the *MRM2* gene did not show any difference in growth at 30 °C compared with the wild type isogenic strain regardless of whether the carbon source is fermentable or non-fermentable. However, growth was severely reduced on a medium supplemented with glycerol at 37 °C, suggesting that the methylation catalyzed by Mrm2 is critical for a proper mitochondrial translation at high temperatures [108].

To study the effects of the p.Gly189Arg mutation identified in humans, the yeast *Saccharomyces cerevisiae* was exploited [116]. The *MRM2* wild type allele and the *mrm2^G259R^* mutant allele, harboring the mutation corresponding to the human one, were cloned in a centromeric vector under its promoter and inserted in a strain deleted in the endogenous *MRM2* gene, which showed a strong decrease in the oxygen consumption rate. Contrarywise to the wild type allele, the mutant allele failed to restore the oxygen consumption rate, indicating that the mutant protein is unable to complement the respiratory phenotype. It was also studied to examine if the mutant allele determines a reduction in Um2791 modification of the yeast 21S mt-rRNA, which corresponds to Um1369 in human 16S mt-rRNA [109]. The ability of the wild type and mutant variants to methylate U2791 was measured through a reverse transcription primer extension assay (RT-PEx), which is based on the retrotranscription of the 21S mt-rRNA using a radioactively labeled primer which anneals downstream from the nucleotide 2791. If the U2791 is methylated, the reverse transcriptase cannot pair a nucleotide and pause at this site, resulting in the blocking of the synthesis of the cDNA and thus in a shorter product. RT-PEx reactions performed on the 21S mt-rRNA from the *mrm2* null strain demonstrated extremely low 2′-O-methyl pausing at U2791, consistent with the lack of modification at this site due to the lack of Mrm2. The expression of a wild type *MRM2* allele restored the Um2791 methylation, whereas the expression of the *mrm2^G259R^* mutant allele resulted in a 30% decrease in the methylated 21S mt-RNA levels compared with the former strain. Analysis in yeast thus confirmed the pathogenicity of the mutant variant, which is associated with a reduction of the methylation activity of the mt-rRNA and a consequent impairment of the oxidative phosphorylation [116].

## 3. Conclusions

Primary mitochondrial disorders (MDs) are a group of clinically heterogeneous defects caused directly or indirectly by an energy impairment. MDs can be caused by mutations in either the nuclear or mitochondrial genome, with a collective incidence of 1.6 in 5000; however, given their high genetic heterogeneity, each single gene defect is extremely rare [214,215].

Among MDs, a particular class is represented by modopathies [6], caused by defective modifications of mt-tRNA or mt-rRNA thus impairing mitochondrial translation. In fact, although the vast majority of mitochondrial proteins are nucleus-encoded, produced in the cytoplasm and imported into mitochondria, 13 subunits of the respiratory complexes in humans are encoded by mtDNA and directly produced through mitochondrial protein synthesis machinery inside the organelle. As a consequence, defective mitochondrial translation results in OXPHOS defects. However, it must be underlined that the clinical phenotypes associated with modopathies and, in general, with several mitochondrial diseases, are not only due to a defect in the OXPHOS activity, but may reside in a general dysfunction of mitochondria and, in some cases, in the toxicity acquired by dysfunctional mitochondria. For example, dysfunctional mitochondria can release their content, among which are mtDNA, modified or not modified mtRNAs, formylated Met-tRNA and unprocessed and formylated peptides, which are able to stimulate the immune system with deleterious consequences such as inflammation [216,217,218].

A large variety of modifications can occur on tRNAs and at least 18 types have been reported for human mt-tRNA (Table 1). tRNA modifications have several functions among which are stabilizing the tRNA structure, avoiding tRNA degradation, avoiding mismatch pairing with the wrong codons, stabilizing codon–anticodon pairing and maintaining the correct reading frame. Considering the key role of these modifications, it is not surprising that mutations in genes involved in these pathways have been associated with pathologies. To date, 10 genes involved in mt-tRNA modifications have been identified as responsible for modopathies (Table 2). Every specific disease has different symptomatology, tissue specificity, age of onset, inheritance and prognosis [6,7,13,59,60,61,62]. Mitochondrial rRNAs are also modified in order to improve rRNA maturation, stability and the assembly of ribosomal subunits and to optimize translation [13,59,105,111,113]. Modifications in 10 positions have been reported on human 12S and 16S mt-rRNAs [100,101]. To date, only one MD is associated with a gene coding for an rRNA modifier: *MRM2* [116].

Most of the mutations in genes coding for mt-tRNA or mt-rRNA modifiers known to date have been identified thanks to next generation sequencing techniques (NGS), encompassing panel sequencing, whole exome sequencing (WES) and whole genome sequencing (WGS). The use of NGS has been fundamental for achieving a genetic diagnosis of several mitochondrial diseases because MDs are amongst the most genetically and phenotypically diverse groups of inherited diseases and suffer from the lack of a single reliable biomarker [215]. The overarching issue of all sequencing approaches is the identification of variants of uncertain significance (VUS). Furthermore, the familiar history is usually not known, and, besides the pathological mutations, other mutations are often identified, thus it is difficult to unequivocally assign a pathological role to the identified variant. For this reason, a functional validation, i.e., to confirm that the variant is the cause of the disorder and not a single nucleotide polymorphism (SNP), is often necessary.

Thanks to its ability to survive and grow when the mitochondrial metabolism is impaired, the yeast *Saccharomyces cerevisiae* has been extensively used to study mitochondrial diseases and has proved its usefulness in validating the pathogenicity of newly identified variants in several genes and in assessing the mechanisms involved in mitochondrial dysfunction. Yeast, therefore, represents a powerful tool to deepen the knowledge about mitochondrial pathways both in health and disease. More generally, the success of this experimental single-cell organism relies on its efficient homologous recombination, which easily and quickly allows the creation of genetic knockouts, and on the easiness of its manipulation thanks to well-known genetic tools, including transformation and plasmid shuffling. Another particularly useful feature is that yeast exists in both haploid and diploid forms. The study of a mutant variant in a haploid genetic background allows it to be immediately validated even if it is recessive in humans. In addition, several intracellular pathways are highly conserved between yeast and humans, including mitochondrial ones. Remarkably, more than 50% of the 1000 proteins estimated to be in yeast mitochondria have a human homolog, and 70% of the nuclear genes involved in human mitochondrial diseases are conserved in yeast [124]. Notably, mutations can be modeled in the yeast *Saccharomyces cerevisiae* without the need for a patient-derived tissue or cell line. One of the most valuable features of using yeast to study the effect of putative pathogenic variants is that it is possible to introduce every allele under investigation in a genetically identical background thus avoiding the possible confounding effects due to the genetic heterogenicity of the patients that could influence the severity and the progression of the disease. It is also possible to study separately two alleles found in compound in a patient in order to determine the contribution of each of them to the phenotype. In addition, the severity of the mitochondrial function impairment observed in yeast often correlates with the severity of the clinical phenotype of the patients.

Although in the last few years the genetic diagnosis of MDs has been improved thanks to the extensive use of NGS, in a large fraction of patients the genetic basis is still unknown. Considering that the human mitochondrial proteome consists of approximately 1500 proteins, it is expected that many more disease genes will be discovered in the coming years. Some functions fulfilled by human mitochondria do not exist in yeast; however, *Saccharomyces cerevisiae* will continue to prove to be an invaluable model system both to prove the pathogenicity of new variants and to dissect the molecular mechanisms underpinning these diseases. Although yeast, as a unicellular organism, cannot help to elucidate tissue-specific defects or phenotypes related to cell-to-cell communication or to immune response, nor for studying the effect of alleged pathological mutations at the level of tissues or organs, it can be exploited to point out the intracellular pathways involved in the disease.

Despite some differences in mitochondrial DNA length and codon usage, a high degree of similarity is shared between yeast and human mtDNA transcription and translation. Several of the mt-RNA modifying enzymes found in humans are present and conserved in yeast, especially the enzymes which modify the tRNAs inside or near the anticodon, thus allowing the use of yeast as a model system. For the non-conserved genes/pathways, other models must be used.

To date, few mutations associated with modopathies were modeled in yeast and the studies designed until now are mostly limited to the validation based on oxidative growth defect and the reduced oxygen consumption rate of the mutant carrying the alleged pathological mutation. In some cases, the molecular consequences were assessed by performing mitochondrial protein synthesis or measuring the levels of the modified RNA. However, it is expected that the number of novel mutations, the number of novel genes and the number of novel modopathies will increase, and the role of yeast in determining a genotype–phenotype correlation and the defects at the molecular level will be likely fundamental. However, it must be underlined that the mitochondrial translation machinery is partially different between humans and yeast, and this could limit, in some cases, the usefulness of yeast for validation. First of all, the 5S rRNA is not imported in yeast mitochondria, so the effects of mutations in such molecules or in proteins involved in its import cannot be studied in yeast. In addition, the number of the proteins of the mitoribosomes and the structures of some of them are different between the two organisms, both in the mtSSU and in the mtLSU, where, for example, the proteins which bind to the tRNA^Val^ are not present in yeast [219,220]. This lack or presence of extra proteins results in the different tridimensional structures of yeast and human mitoribosomes [219]. Finally, ribosome steady-state levels are finely regulated by specific assembly factors and proteases, some of which, such as the caseinolytic mitochondrial matrix peptidase proteolytic subunit (CLPP), are present in humans but not in *Saccharomyces cerevisiae* [221,222].

Despite the tremendous progress made in MD’s diagnosis over the last years, therapy is still insufficient and mostly limited to relieving symptoms. Therefore, there is an urgent need to develop new drugs to treat these diseases, especially pathologies recently discovered, such as most modopathies. In order to identify drugs able to rescue the detrimental effects of pathological mutations, a high-throughput phenotype-based approach called “drug drop test” was set up in yeast allowing to quickly screen a large number of molecules taking advantage of the oxidative defect of the strains defective in mitochondrial functions [123]. In the last few years, this method was successfully used on a variety of mitochondrial disease models leading to the identification of potential therapeutic molecules [223,224,225,226,227,228,229,230]. Notably, some of these compounds that were identified as effective in yeast have also shown promising results in the corresponding animal models and in patients’ fibroblasts [224,229,231]. Since, in many cases, pathological mutations in tRNA or rRNA modifying enzymes are not associated with a complete loss of oxidative growth, as reported above, it will be possible to exploit these models to perform a drug library screening in an attempt to identify beneficial molecules. Furthermore, it could be interesting to address this question in models of different modopathies in order to assess if any drugs could have a broad-spectrum beneficial effect on mitochondrial translation.

## Figures and Tables

**Figure 1 ijms-24-02178-f001:**
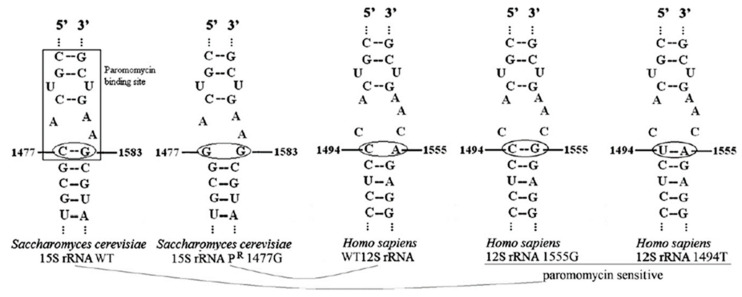
Secondary structure of the A-site of the wild type and P^R^ yeast strain 15S mt-rRNA, and of the wild type and aminoglycoside-sensitive human 12S mt-rRNA, as reported in [69].

**Figure 2 ijms-24-02178-f002:**
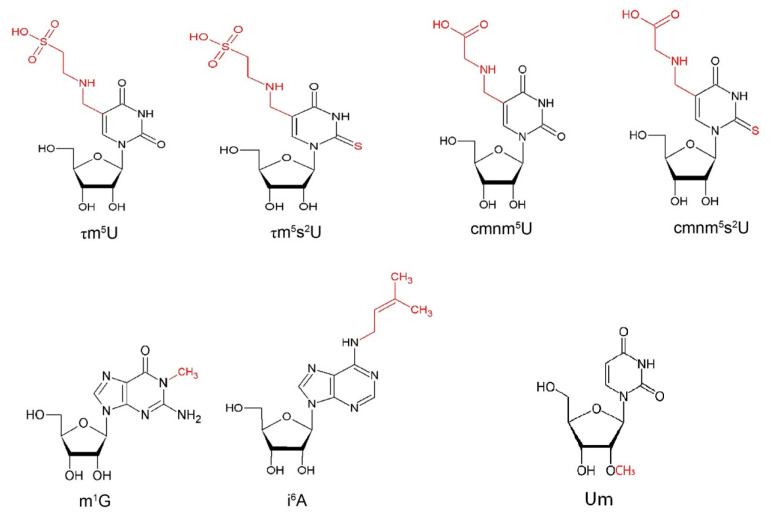
Modified nucleotides found in human and yeast mt-tRNAs and mt-rRNAs after the modifications catalyzed by the enzymes reported in Section 2. Modifications are highlighted in red. Abbreviations are as in Table 1, Table 2, Table 3 and Table 4.

**Table 1 ijms-24-02178-t001:** Nucleotide modifications found in human mt-tRNAs, position of the modification, human gene(s) encoding for the modifying enzyme(s) and their yeast orthologs, if present.

Modification	Position	Human Modifying Enzyme Gene	Yeast Ortholog	References
G adding at 5′		*THG1L*?	*THG1*	[8,9]
m^1^G	9	*TRMT10C*, *HSD17B1*	*TRM10*,-	[10,11]
m^1^A	9	*TRMT10C*, *HSD17B1*	*TRM10*,-	[10]
m^2^G	10	*TRMT11/TRMT112?*	*TRM11*/*TRM112*	[12]
m^1^A	14	*TRM10TC?*	*TRM10*	[13]
m^1^A	16	*TRM10TC*?	*TRM10*	[13]
D	20	Unknown (*DUS2L?*)	*SMM1*	[14,15]
m^2^G, m^2^_2_G	26	*TRMT1*	*TRM1*	[16,17]
Ψ	27,28	*PUS1*	*PUS1*/*PUS2*	[18,19]
Ψ	31	*RPSUD1*/*RPUSD2*?	*PUS6* ^a^	[20]
m^3^C	32	*METTL2A/2B/6/8?*	*ABP140*(*TRM140*)	[21]
Ψ	32	*RPSUD1*/*RPUSD2*?	*PUS9* ^a^	[22]
Ψ	33	*RPSUD1*/*RPUSD2*?	-	[21]
τm^5^U	34 ^b^	*MTO1*, *GTPBP3*	*MTO1*/*MSS1*	[6,23,24,25,26,27]
τm^5^s^2^U	34 ^b^	τm^5^U+*TRMU*(*MTU1*)	*SLM3*(*MTO2*, *MTU1*)	[28,29]
f^5^C	34 ^b^	*NSUN3*, *ALKBH1*	-	[30,31,32]
Q	34 ^b^	*QTRT1*, *QTRT2*	-	[21,33]
Ψ	35 ^b^	*RPSUD1*/*RPUSD2*?	-	[21]
m^1^G	37	*TRMT5*	*TRM5*	[34,35]
t^6^A	37	*YRDC*, *OSGEPL1*	*SUA5*, *QRI7*	[36,37,38,39]
i^6^A	37	*TRIT1*	*MOD5*	[40,41,42]
ms^2^i^6^A	37	i^6^A + *CDK5RAP1*	-	[43,44]
Ψ	38	*RPSUD1*/*RPUSD2*?	-	[21]
Ψ	39	*RPUSD4*	-	[45]
Ψ	40	*RPSUD1*/*RPUSD2*?	-	[21]
m^5^C	48,49,50	*NSUN2*	*NCL1*	[46,47]
Ψ	50	*RPSUD1*/*RPUSD2*?	-	[21]
m^5^U	54	*TRMT2B*	*TRM2*	[48,49]
Ψ	55	*TRUB2*?	*PUS4*	[50,51]
m^1^A	58	*TRMT61B*	*GCD14*	[52,53]
Ψ	66,67,68	*RPSUD1*/*RPUSD2*?	-	[21]
CCA adding at 3′		*TRNT1*	*CCA1*	[54,55]

m^1^G: 1-methylguanosine; m^1^A: 1-methyladenosine; m^2^G; N2-methylguanosine; D: dihydrouridine; m^2^_2_G: N2-dimethylguanosine; Ψ: pseudouridine; m^3^C: 3-methylcytidine; τm^5^U: 5-taurinomethyluridine; τm^5^s^2^U: 5-taurinomethyl-2-thiouridine; f^5^C: 5-formylcytidine; Q: queuosine; t^6^A: N6-threonylcarbamoyladenosine; i^6^A: N6-isopentenyladenosine; ms^2^i^6^A: 2-methylthio-N6-isopentenyladenosine; m^5^C: 5-methylcytidine; m^5^U: 5-methyluridine. ^a^ *PUS6* and *PUS9* encode for enzymes catalyzing the modification in yeast, but it is not known if the most similar human genes have the same role in the corresponding human mt-tRNA modification. ^b^ Positions 34 and 35 correspond to the first and second nucleotides of the anticodon, respectively.

**Table 2 ijms-24-02178-t002:** Genes encoding for mt-tRNA modifying enzymes associated with mitochondrial diseases. Yeast studies are reported only if yeast was used to confirm the pathogenicity of the mutations found in patients, or in general for studying mutations affecting the efficiency of modification. Abbreviations: HCLA: hypertrophic cardiomyopathy and lactic acidosis; MELAS: mitochondrial encephalomyopathy, lactic acidosis and stroke-like episodes; MERRF: myoclonic epilepsy with ragged red fibers; HCLA: hypertrophic cardiomyopathy and lactic acidosis; RIRCD: reversible infantile respiratory chain deficiency.

Human Gene	Protein Function	Disease	OMIM Number	Onset	Inheritance	Main Phenotype	References	Yeast Gene	Study in Yeast
*TRMT10C*	Methylation of A to m^1^A and G to m^2^G at position 9	Combined oxidative phosphorylation deficiency 30	616974	Infancy	AR	Microcephaly, cerebellar ataxia, Intellectual disability, nephropathy, short stature	[63]	*-*	-
*HSD17B10* ^a^	Methylation of A to m^1^A and G to m^2^G at position 9	HSD10 mitochondrial disease	300438	Childhood	XR/XD	Neurodegeneration, cardiomyopathy, early death	[64]	-	-
*PUS1*	Pseudouridylation at positions 27 and 28	Myopathy, lactic acidosis and sideroblastic anemia 1	600462	Childhood	AR	Mitochondrial myopathy, sideroblastic anemia (MLASA)	[18,65,66,67,68]	*PUS2*	-
*MTO1*	Taurinomethylation of U to τm^5^U at position 34	Combined oxidative phosphorylation deficiency 10 with MELAS, MERRF, HCLA and/or RIRCD	614702	Infancy, childhood	AR	Hypertrophic cardiomyopathy, lactic acidosis, ID, short stature, early death	[69,70,71,72,73,74]	*MTO1*	[69,70]
		Maternally inherited deafness		Different age	Mit, Phenotypic modifier	Deafness in presence of specific mtDNA affecting 12S mt-rRNA	[23,75,76]		
*GTPBP3*	Taurinomethylation of U to τm^5^U at position 34	Combined oxidative phosphorylation deficiency 23 with MELAS, MERRF, HCLA and/or RIRCD	616198	Infancy, childhood	AR	Intellectual disability, hearing loss, short stature, early death	[77,78]	*MSS1*	[25]
		Maternally inherited deafness		Different age	Mit, Phenotypic modifier	Deafness in presence of specific mtDNA affecting 12S mt-rRNA	[25,75]		
*TRMU*	Thiolation of τm5U to τm5s2U at position 34	Liver failure, transient infantile	613070	Infancy	AR	Hepatopathy, lactic acidosis, Leigh syndrome, hearing loss, early death	[79,80,81,82,83,84]	*MTO2*	[28,29]
		Maternally inherited deafness, aminoglycoside-induced	580000	Different age	Mit, Phenotypic modifier	Deafness in presence of specific mtDNA affecting 12S mt-rRNA	[28,85,86]		
*NSUN3*	Methylation of C followed by oxidation by ALKBH1 to f^5^C at position 34	Combined oxidative phosphorylation deficiency 48	619012	Infancy	AR	Microcephaly, seizure, lactic acidosis, muscle weakness, short stature, failure to thrive, external ophthalmoplegia, convergence nystagmus	[30,87]	*-*	-
*TRMT5*	Methylation of G to m^1^G at position 37	Peripheral neuropathy with variable spasticity, exercise intolerance and developmental delay	616539	Infancy, childhood, young adulthood	AR	Cardiomyopathy, lactic acidosis, demyelinating neuropathy, renal tubulopathy, cirrhosis, short stature	[88,89,90]	*TRM5*	[88]
*TRIT1*	Isopenthynilation of A to i^6^A at position 37	Combined oxidative phosphorylation deficiency 35	617873	Infancy	AR	Microcephaly, intellectual disability, cardiomyopathy, encephalopathy, myoclonic epilepsy	[41,91]	*MOD5*	[41]
*TRNT1* ^b^	Addition of CCA to the tRNA 3′	Sideroblastic anemia with B-cell immunodeficiency, periodic fevers and developmental delay	616084	Infancy	AR	Sideroblastic anemia with immunodeficiency, fevers, developmental delay (SIFD)	[54,92]	*CCA1*	[54,93]

^a^ *HSD17B10* encodes for an enzyme with multiple functions. Here, only pathologies associated with a defect of the methyltransferase activity and the relative literature are reported. ^b^ *TRNT1* encodes for a nucleotidyltransferase that adds CCA stretch to the 3′ of both cytoplasmic and mt-tRNAs. The diseases reported and the relative literature is limited to pathologies for which a clear involvement of mitochondrial defects is reported.

**Table 3 ijms-24-02178-t003:** Nucleotide modifications found in human mt-rRNAs, the position of the modification (referred to the mtDNA encoding nucleotide), human gene(s) encoding for the modifying enzyme(s) and their yeast orthologs, if present.

Modification	Position	Human Modifying Enzyme Gene	Yeast Ortholog	References
m^5^U	429 (12S)	*TRMT2B*	*TRM2* ^a^	[48,103,104]
m^4^C	839 (12S)	*METTL15*	*-*	[103,104]
m^5^C	841 (12S)	*NSUN4*	*-*	[105]
m^6^_2_A	936 (12S)	*TFB1M*	*-*	[106]
m^6^_2_A	937 (12S)	*TFB1M*	*-*	[106]
m^1^A	947 (16S)	*TRMT61B*	*GCD14* ^a^	[101]
Gm	1145 (16S)	*MRM1*	*MRM1*	[107]
Um	1369 (16S)	*MRM2*	*MRM2*	[107,108,109]
Gm	1370 (16S)	*RNMTL1*(*MRM3*)	*-*	[107,109]
Ψ	1397 (16S)	*RPUSD4*	*PUS5*	[45,110]

m^5^U: 5-methyluridine; m^4^C: N4-methylcytidine; m^5^C: 5-methylcytidine; m^6^_2_A: N6,N6-dimethyladenosine; m^1^A: 1-methyladenosine; Gm: 2′-O-methylguanosine; Um: 2′-O-methyluridine; Ψ: pseudouridine. ^a^ Yeast *TRM2* and *GCD14* seem to be involved only in the modifications of tRNAs.

**Table 4 ijms-24-02178-t004:** Human mt-tRNAs with their anticodon with modifications according to [21], and their yeast corresponding mt-tRNAs with their anticodon with modifications according to [28], and the recognized codons; human mt-rRNAs and their corresponding yeast mt-rRNAs.

Human mt-RNA	Yeast mt-RNA	Codons Recognized
tRNA^Ala(UGC)^	tRNA^Ala(UGC)^	GCN
tRNA^Arg(UCG)^	tRNA^Arg(ACG)^	CGN
-	tRNA^Arg(UCU)^	AGR (in yeast)
tRNA^Asn(QUU)^	tRNA^Asn(GUU)^	AAY
tRNA^Asp(QUC)^	tRNA^Asp(GUC)^	GAY
tRNA^Cys(GCA)^	tRNA^Cys(GCA)^	UGY
tRNA^Gln(τm5s2UUG)^	tRNA^Gln(cmnm5s2UUG)^	CAR
tRNA^Glu(τm5s2UUC)^	tRNA^Glu(cmnm5s2UUC)^	GAR
tRNAGly^(UCC)^	tRNA^Gly(UCC)^	GGN
tRNAHis^(QUG)^	tRNA^His(GUG)^	CAY
tRNAIle^(GAU)^	tRNA^Ile(GAU)^	AUY
tRNA^Leu(UAG)^	-	CUN (in humans)
tRNA^Leu(τm5UAA)^	tRNA^Leu(UAA)^	UUR
tRNA^Lys(τm5s2UUU)^	tRNA^Lys(cmam5s2UUU)^	AAR
tRNA^(f)Met(f5CAU) a^	tRNA^fMet(CAU) a^	AUR
	tRNA^Met(CAU) a^	
tRNA^Phe(GAA)^	tRNA^Phe(GAA)^	UUY
tRNA^Pro(UGG)^	tRNA^Pro(UGG)^	CCN
tRNA^Ser(GCU)^	tRNA^Ser(GCU)^	AGY
tRNA^Ser(UGA)^	tRNA^Ser(UGA)^	UCN
tRNA^Thr(UGU)^	tRNA^Thr(UGU)^	ACN
-	tRNA^Thr(UAG)^	CUN (in yeast)
tRNATrp^(τm5UCA)^	tRNA^Trp(UCA)^	UGR
tRNATyr^(QUA)^	tRNA^Tyr(GUA)^	UAY
tRNAVal^(UAC)^	tRNA^Val(UAC)^	GUN
12S rRNA	15S rRNA	-
16S rRNA	21S rRNA	-

Abbreviations are as in Table 1. Other abbreviations: cmcm^5^s^2^U: 5- 5-carboxymethylaminomethyl -2-thiouridine; Y: pyrimidine; R: purine; N: every nucleotide; fMet: formyl-methionine. ^a^ Humans possess a single tRNA for formyl-methionine and methionine, yeast possesses a tRNA for formyl-methionine (initiator tRNA) and a tRNA for methionine.

## Data Availability

Not applicable.

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
