# Peer review of "Modopathies Caused by Mutations in Genes Encoding for Mitochondrial RNA Modifying Enzymes: Molecular Mechanisms and Yeast Disease Models"

_ijms, 2023, doi:10.3390/ijms24032178_

Round 1

Reviewer 1 Report

In the review, "Modopathies caused by mutations in genes encoding for mitochondrial RNA modifying enzymes: molecular mechanisms and yeast disease models" Magistrati et al. organize the literature to ascertain a class of genes involved in processing mitochondrial RNA. The review is well-written, updated, and relevant for researchers studying mitochondria diseases. The proper function in tRNAs and rRNAs base modifications is didactic and included in tables. The unique criticism that I may have for this review is the absence of any mention of other proteins that modify mitochondrial RNAs such as RNase P, RNase Z, and FMT1. To be fair, RNase P is mentioned en passant, but not explored. If the argument to justify their absence was the focus on those involved in base modification, why CCA1 was included?  

Minor corrections/comments:

In lines 171, 172 and so on the yeast gene's names are not in italic. 

Line 460 - For clarity's sake, they should indicate what kind of mtDNA background was first tested on the mto1 null mutant - probably Paramomycin sensitive as we can interpret from what comes next. 

Line 491 - I believe the authors refer to the newly synthesized accumulation of the mentioned mitochondrial products, not the steady state level as the text implies. 

Lines 578-585 - The description of TRM5 function on mitochondrial and cytosolic RNAs is somewhat confusing. In the description of Mod5 (668-675) it is much better defined the protein function in each cell compartment. 

Line 770 - I would remove the adjective "strong" to qualify the ADH promoter. 

Author Response

In the review, "Modopathies caused by mutations in genes encoding for mitochondrial RNA modifying enzymes: molecular mechanisms and yeast disease models" Magistrati et al. organize the literature to ascertain a class of genes involved in processing mitochondrial RNA. The review is well-written, updated, and relevant for researchers studying mitochondria diseases. The proper function in tRNAs and rRNAs base modifications is didactic and included in tables.

We thank the reviewer for the comments and the suggestions, in particular those concerning the discussion of CCA1. Here enclosed our replies.

The unique criticism that I may have for this review is the absence of any mention of other proteins that modify mitochondrial RNAs such as RNase P, RNase Z, and FMT1. To be fair, RNase P is mentioned en passant, but not explored. If the argument to justify their absence was the focus on those involved in base modification, why CCA1 was included?  

Concerning the inclusion of mutations in genes encoding for RNases, we decided not to include them since this processing, strictly speaking, does not modify nucleotides (as you underlined), and we think that processing of pre-RNA would require an ad hoc review, due also to the amount of information linked to these few genes. Concerning FMT1, we did not include it since it modifies an amino acid (however, strictly speaking, the methionine is linked to a nucleotide of the tRNA, so it is a modification) rather than directly a nucleotide. Finally, we included CCA1 since the addition of the CCA stretch at 3’, as well as the addition of a single G at 5’ of some tRNAs is critical for the activity of tRNAs, and some authors include these enzymes among post-transcriptional tRNA modifying enzymes and the mutations in these genes as causing modopathies. We agree, however, that it is not a nucleotide modification of the tRNAs, and we amended the text. (Lines 742-745 in the manuscript with the track changes)

Minor corrections/comments:

In lines 171, 172 and so on the yeast gene's names are not in italic.

Done (Lines 181-182)

Line 460 - For clarity's sake, they should indicate what kind of mtDNA background was first tested on the mto1 null mutant - probably Paramomycin sensitive as we can interpret from what comes next.

Done (Lines 473-474)

Line 491 - I believe the authors refer to the newly synthesized accumulation of the mentioned mitochondrial products, not the steady state level as the text implies.

Correct. We amended the text. (Lines 501-506)

Lines 578-585 - The description of TRM5 function on mitochondrial and cytosolic RNAs is somewhat confusing. In the description of Mod5 (668-675) it is much better defined the protein function in each cell compartment.

We tried to better distinguish the role of cytoplasmic Trm5 and mitochondrial Trm5, focusing in particular on the methylation of the inosine at position 37 and on the participation in the formation of wybutosine at position 37 in two specifics cytoplasmic tRNAs. (Lines 593-603)

Line 770 - I would remove the adjective "strong" to qualify the ADH promoter.

Done (Line 807)

Reviewer 2 Report

The authors have a tradition to write quality reviews about mitochondriopathies, and have added a valuable manuscript now, focusing on "modopathies". Their compilation of literature is almost perfect, their personal interpretations and cautions are well balanced, and the English usage is almost faultless. Overall, it was a nice update for me to read this paper and it will benefit many other researchers. However, the authors seem unaware of other recent developments, so they might want to discuss additional references.

Minor criticisms:

(1) In the first sentence of Conclusions, the statement "mitochondriopathies due to energy impairment" paints a very simple picture. If the loss of energy were the single crucial issue, then the phenotype variability of mitochondriopathies and modopathies could not be explained. The authors ignore excellent evidence that dysfunctional mitochondria acquire toxicity, e.g. due to the release of hypomethylated mtDNA / mtRNA / formylated peptides / metabolits, which activate the innate immune system and have deleterious consequences. Therefore, the action of cytosolic enzymes that enter mitochondria to methylate RNA may also function to minimize toxicity of mtRNA in eukaryotic organisms. Relevant literature perhaps PubMed-ID 21799518, 35223833.

(2) The authors also ignore other roles of mitochondria e.g. for apoptosis, calcium buffering, lipid metabolism, polyamine synthesis, and iron homeostasis. Particularly iron/heme/[Fe-S]clusters are important for RNA/DNA stability also in mitochondria, polyamines are tRNA conformation stabilizers also in mitochondria. Relevant literature perhaps PubMed-ID 31834895, 23933583, 29967381, 32817343, 22678362; and 30347855, 19962967. The authors should either include such literature or clarify that their review focuses only on the covalent modifications of mitochondrial RNA.

(3) Although the advantages of yeast to understand mutation effects are highlighted in detail, the big differences between yeast and human regarding the translation apparatus are not mentioned: Many mitoribosomal large subunit tRNA-Val binding proteins are not conserved from human to yeast (PMID 25278503). Mitochondrial proteostasis during stress is very different in yeast given that CLPP (as one of two matrix proteases and as mitoribosome modulator) does not exist there (PMID 33073672). The important differences between human rRNA and yeast rRNA seem to be ignored by the authors, given that in their previous review Magistrati et al 2022 IJMS they stated that only tRNA-Val/Phe but not 5S-rRNA is present in mitoribosomes, apparently unaware of recent findings that 5S-rRNA can be recruited to the mitoribosome in stress periods (PMID 21685364, 24120868, 25866880).

Author Response

The authors have a tradition to write quality reviews about mitochondriopathies, and have added a valuable manuscript now, focusing on "modopathies". Their compilation of literature is almost perfect, their personal interpretations and cautions are well balanced, and the English usage is almost faultless. Overall, it was a nice update for me to read this paper and it will benefit many other researchers. However, the authors seem unaware of other recent developments, so they might want to discuss additional references.

We thank the reviewer for the comments and the suggestions, in particular those regarding the differences between humans and yeast and the non-covalent interaction with nucleic acids.

Minor criticisms:

(1) In the first sentence of Conclusions, the statement "mitochondriopathies due to energy impairment" paints a very simple picture. If the loss of energy were the single crucial issue, then the phenotype variability of mitochondriopathies and modopathies could not be explained. The authors ignore excellent evidence that dysfunctional mitochondria acquire toxicity, e.g. due to the release of hypomethylated mtDNA / mtRNA / formylated peptides / metabolits, which activate the innate immune system and have deleterious consequences. Therefore, the action of cytosolic enzymes that enter mitochondria to methylate RNA may also function to minimize toxicity of mtRNA in eukaryotic organisms. Relevant literature perhaps PubMed-ID 21799518, 35223833.

We strongly agree that the phenotypes associated with mutations in genes associated with mitochondrial diseases, including modopathies, may be caused by effects not linked directly to OXPHOS defects, such as inflammation. Although not the focus of this review, due to the importance of this argument, we added a small subparagraph at the beginning of the conclusion, citing the literature suggested by the reviewer and some additional literature. We thank the reviewer for this information, which we will deepen surely. (Lines 942-949 in the manuscript with the track changes)

(2) The authors also ignore other roles of mitochondria e.g. for apoptosis, calcium buffering, lipid metabolism, polyamine synthesis, and iron homeostasis. Particularly iron/heme/[Fe-S]clusters are important for RNA/DNA stability also in mitochondria, polyamines are tRNA conformation stabilizers also in mitochondria. Relevant literature perhaps PubMed-ID 31834895, 23933583, 29967381, 32817343, 22678362; and 30347855, 19962967. The authors should either include such literature or clarify that their review focuses only on the covalent modifications of mitochondrial RNA.

We ignore this aspect in the review since, as you indicated, we focused on covalent modifications in mtRNAs. However, again, we added in the introduction that mtRNA can interact with other molecules in a non-covalent way and the consequences of these interactions, citing some of the literature suggested. (lines 155-160)

(3) Although the advantages of yeast to understand mutation effects are highlighted in detail, the big differences between yeast and human regarding the translation apparatus are not mentioned: Many mitoribosomal large subunit tRNA-Val binding proteins are not conserved from human to yeast (PMID 25278503). Mitochondrial proteostasis during stress is very different in yeast given that CLPP (as one of two matrix proteases and as mitoribosome modulator) does not exist there (PMID 33073672). The important differences between human rRNA and yeast rRNA seem to be ignored by the authors, given that in their previous review Magistrati et al 2022 IJMS they stated that only tRNA-Val/Phe but not 5S-rRNA is present in mitoribosomes, apparently unaware of recent findings that 5S-rRNA can be recruited to the mitoribosome in stress periods (PMID 21685364, 24120868, 25866880).

We agree that there are some significant differences, so we amended the text in the introduction and in the conclusions. Concerning the 5S rRNA, we are aware that, at least in mammals, they are imported into mitochondria, but we did not focus on that since 5S RNA is not modified (at least in mammals, not in yeast). However, rethinking the review aims, we think that this aspect must be indicated, as we now did in the introduction. (Lines 133-137).

We underlined the difference between mitochondrial apparatus in yeast and mammals as a limit in the conclusions (Lines 1026-1038)